# Semantic Processing Deficits and Their Use as Early Biomarkers in Schizophrenia

**DOI:** 10.3390/healthcare13161958

**Published:** 2025-08-10

**Authors:** Alfonso Martínez-Cano, Begoña Polonio-López, Juan José Bernal-Jiménez, José L. Martín-Conty, Laura Mordillo-Mateos, Manuela Martinez-Lorca

**Affiliations:** 1Faculty of Health Sciences, University of Castilla-La Mancha, 45600 Talavera de la Reina, Spain; alfonso.martinez@uclm.es (A.M.-C.); laura.mordillo@uclm.es (L.M.-M.); 2Department of Medical Sciences, University of Castilla-La Mancha, 02071 Albacete, Spain; 3Department of Occupational Therapy Physiotherapy and Nursing, Technological Innovation Applied to Health Research Group (ITAS Group), University of de Castilla-La Mancha, 45600 Talavera de la Reina, Spain; 4Department of Nursing, Physiotherapy and Occupational Therapy, Faculty of Health Sciences, Universidad de Castilla-La Mancha, 45600 Talavera de la Reina, Spain; juanjose.bernal@uclm.es (J.J.B.-J.); joseluis.martinconty@uclm.es (J.L.M.-C.); 5Faculty of Health Sciences, Department of Psychology, University of Castilla-La Mancha, 02071 Albacete, Spain; manuela.martinez@uclm.es

**Keywords:** schizophrenia, semantic processing deficits, high risk, language

## Abstract

**Background**: Schizophrenia is a serious mental health condition that usually begins in adolescence and often progresses to become a chronic and disabling illness. Difficulties in communication and anomalous language are considered core elements of the disorder. Several studies have demonstrated the presence of semantic deficits in individuals with schizophrenia, suggesting that these deficits may constitute a core feature of the disorder. However, research in this area remains limited, particularly among individuals at high risk of developing the disorder. The central hypothesis of this study is that individuals with schizophrenia exhibit semantic processing deficits, even when cognitive function, psychopathology, and medication are controlled for. We also hypothesize that similar, albeit milder, deficits can be observed in individuals at high risk of developing the condition. **Methods**: This cross-sectional study included 155 participants divided into three groups: 46 with schizophrenia, 42 at high risk due to factors like substance use and high psychopathology, and 67 controls matched by sex, age, and education. Semantic processing was assessed using the semantic relations subtest from the BETA, controlling for medication and cognitive performance as possible confounding factors. **Results**: the results revealed significant differences among the three groups (F = 28.543; *p* < 0.001); the schizophrenia group performed poorly, followed by the high-risk group, and then the control group, which showed no deficits. Error patterns were also analyzed to assess group differences, revealing that the schizophrenia group had the lowest scores and the most specific deficits. These findings highlight the relevance of semantic evaluation in schizophrenia and, more importantly, in individuals at high risk of developing the disorder, as such deficits may serve as early biomarkers. Additionally, significant correlations were found between semantic performance and variables such as medication (r = −0.342; *p* = 0.020), cognition (r = −0.259; *p* = 0.001), and psychopathology (r = −0.566; *p* < 0.001). **Conclusions**: This emphasizes the need to control these factors to avoid misinterpreting semantic deficits in both schizophrenia and high-risk groups. The present research is not without limitations; for example, the study design does not allow for conclusions of causality but rather of correlation.

## 1. Introduction

Schizophrenia is a serious mental health condition that typically begins in adolescence and frequently evolves into a chronic and disabling illness [1,2]. Impairments in communication and atypical language use are considered core features of the disorder [3,4].

Research has documented deficits in prosody, speech patterns, and pausing [5], as well as in syntax [6,7] and pragmatic abilities [8,9]. However, semantics and lexical access are arguably the most widely studied aspects of language in schizophrenia [10,11]. Specifically, individuals with schizophrenia tend to show greater impairments in semantic verbal fluency compared to phonological fluency [10,12].

Semantic knowledge has been conceptualized as a structured network of interconnected nodes, whose co-activation gives rise to a conceptual representation [13]. The proximity of these nodes is determined by shared characteristics among concepts, a notion referred to in philology as ’semantic intension’, the set of essential properties or features that define a concept [11,14].

Significant deficits in semantic memory and visual confrontation naming have been observed in individuals with schizophrenia [11,12,15], with action naming being more impaired than object naming [16].

From a clinical perspective, two main types of schizophrenia patient are typically identified based on symptoms: (a) patients with predominantly positive symptoms, which include bizarre and abundant semiology, flight of ideas, and disorganized thought, and (b) patients with predominantly negative symptoms, who tend to present alogia and abulia as primary features [1,17,18].

These symptom profiles are associated with distinct patterns of semantic processing. In patients with positive symptoms, semantic impairments may stem from hyperactivation or aberrant activation of semantic nodes, whereas in patients with negative symptoms, such deficits may be linked to hypoactivation [19,20].

Despite numerous studies having been conducted on semantic deficits in people with schizophrenia, many fail to consider influential factors such as cognition or medication. For instance, low scores on cognitive tests, which are common in these patients, affect language, and greater cognitive impairment has been shown to be associated with greater language deficits [21,22]. Additionally, some studies do not control for the impact of antipsychotics, despite them being associated with slower articulation, more pauses, and shorter sentences [23].

Similar semantic impairments have been identified in high-risk populations; however, existing studies have not established a causal link between these deficits and the eventual onset of schizophrenia. This raises critical questions about the relationship between semantic dysfunction and the development of psychosis, and whether such deficits could serve as early biomarkers [24].

To explore the presence of semantic impairments in at-risk populations, it is essential to distinguish between two key groups: those with Clinical High Risk (CHR), who exhibit subtle and non-specific symptoms, and those classified as Ultra High Risk (UHR), defined by the presence of Attenuated Psychotic Symptoms (APS) or Brief Limited Intermittent Psychotic Symptoms (BLIPS) [25,26,27].

The present study focused on individuals classified as high-risk, particularly those exhibiting basic symptoms, regarded as early prodromal indicators of schizophrenia [28]. These symptoms manifest as subjective disturbances that occur early on and precede the onset of psychosis, affecting functions such as volition, thought, language, and perception [29,30]. Participants were also required to meet additional inclusion criteria linked to increased risk, such as being between 15 and 33 years of age [31,32].

Notably, the sample included individuals exhibiting substance abuse behaviors, which further increases their vulnerability. Several studies have shown that between 70% and 80% of individuals with schizophrenia report high levels of substance use [33,34]. This strong association has been partially explained by the self-medication hypothesis, which posits that substance use may serve as a strategy to alleviate psychopathological symptoms [35,36].

The central hypothesis of this study is that individuals with schizophrenia exhibit semantic processing deficits, even when cognitive function, psychopathology, and medication are controlled for. We also hypothesize that similar, albeit milder, deficits can be observed in individuals at high risk of developing the condition.

Semantic processing is operationalized through the concept of semantic intention [37], defined as the ability to identify the set of defining properties that an entity must meet for a term to apply, and to detect the item that violates these criteria. This capacity was assessed using the Semantic Relations subtest of the BETA Test [38].

Given the lack of standardized linguistic assessments specifically designed for individuals with schizophrenia, this study provides valuable insights into their language deficits and evaluates the potential of such tests for early diagnosis and intervention.

## 2. Methodology

### 2.1. Sample

The present study consisted of a sample of 155 participants, divided into three groups: 46 individuals diagnosed with schizophrenia (SP), 42 individuals at high risk of developing the disorder (HR), and 67 individuals in the control group (CG). The control group was equated with the HR group based on sex, age, and educational level.

The final sample was predominantly male (61.5%), with the majority having completed secondary education (84.9%) and 52.1% reporting a family history of mental disorders in first-degree relatives.

### 2.2. Study Design

This study employed a cross-sectional, non-experimental, and descriptive design. A synchronic (single time-point) approach was used for data collection. To ensure that participants met the inclusion and exclusion criteria, indicated to the collaborating institutions, several assessments were conducted.

The SCIP-S test was administered to rapidly screen cognitive functioning, and all participants scored within the expected range. To rule out substance use disorder as a primary diagnosis, both the frequency and type of substance use were assessed.

The CAPE-42 questionnaire was used to confirm that individuals in the HR group exhibited basic and prepsychotic symptoms associated with an elevated risk of schizophrenia, while the CG did not. As a result, seven participants from the control group (who scored above 2.8 on the positive dimension of the CAPE-42) and two from the high-risk group (who scored below 2.8 on the same dimension) were excluded from the study for not meeting the expected psychopathological criteria.

Once these criteria were verified, all remaining participants completed the semantic relations subtest from the BETA test. Therefore, group allocation was based on diagnostic and psychometric criteria. While cross-sectional designs do not allow for causal inferences, they are well-suited to identifying patterns, correlations, and risk indicators, and can serve as a foundation for future longitudinal research.

By relying on psychometric data, we can be confident that the high-risk group indeed presents the target risk variable in contrast to the control group. Furthermore, the equating of participants on sociodemographic variables such as age, sex, and educational level helps reduce the influence of these factors on the results.

Additionally, the inclusion of baseline measures that ruled out cognitive deficits and substance use disorders further minimizes the potential confounding effects of these variables in the interpretation of the findings.

### 2.3. Procedure

The study was approved by the Research Ethics Committee of the University of Castilla-La Mancha. Participants gave their informed consent either personally or through their legal guardians. The research was conducted in accordance with the latest official version of the World Medical Association’s (WMA) Declaration of Helsinki (1964), as well as the General Council of Official Colleges of Psychologists’ Code of Practice.

People with schizophrenia (PS) were recruited through the psychiatric unit at a Spanish hospital and psychosocial rehabilitation and supervised housing and nursing home services. Inclusion criteria for this group included having a confirmed schizophrenia diagnosis (DSM-V) and having their symptoms assessed using the Positive and Negative Syndrome Scale (PANSS) [39]. A minimum of eighteen months since diagnosis was required for all participants, in order to reduce the likelihood of false positives and ensure the exclusion of comorbid conditions. Individuals with a primary diagnosis other than schizophrenia were excluded from the study, as confirmed by the collaborating psychiatrists.

High-risk (HR) participants were recruited from various Proyecto Hombre facilities (an association for people with drug dependence problems) across Spain. To be included in the HR group, participants had to meet the clinical criteria for APS, as defined in Section III of the DSM-5-TR [40]. They also had to exhibit basic symptoms indicative of a subthreshold psychotic mental state characterized by psychotic-like experiences differing in intensity and frequency from full-blown psychosis [41]. These symptoms were identified based on weighted scores above 2.8 on the positive dimension, according to the criteria proposed by Mossaheb et al. [42]. In their study, when comparing the CAPE-42 with the gold-standard high-risk assessment tool —the Comprehensive Assessment of At-Risk Mental States (CAARMS)—[43], they found that these thresholds yielded higher sensitivity (83%) and a higher negative predictive value (74%), albeit with lower specificity (49%) and positive predictive value (63%). Participants also had to be aged between 15 and 33, in line with the high-risk criteria established by the PACE Clinic [44]. Individuals with a diagnosed substance use disorder or any other mental disorder were excluded from the study.

The control group (CG) consisted of individuals recruited from various secondary schools in Spain. To be included in the group, participants were required not to have a schizophrenia diagnosis and not to meet any of the inclusion criteria set for the HR group. CG participants were matched with their HR group counterparts for sex, age, and level of education, excluding subjects presenting basic symptoms of prodromal schizophrenia as assessed by the CAPE-42. Individuals with weighted scores exceeding 2.8 for the positive symptom dimension were excluded from the study.

Bilingualism was an exclusion criterion for all groups because studies have shown that bilingual individuals have greater semantic fluency than monolingual individuals [45], which could act as a confounding variable.

To recruit the three groups—individuals with schizophrenia, high-risk individuals, and controls—we contacted the hospital’s chief psychiatrist, clinical psychologists from the Proyecto Hombre Association, and school counselors. The inclusion and exclusion criteria were explained to them, and they identified potential participants who met these criteria. We assessed the candidates to confirm eligibility, excluding those who did not meet the established criteria.

First, participants in the high-risk group were recruited based on two criteria: recurrent substance use and being between 15 and 33 years of age. After a psychopathological assessment using the CAPE-42 and confirmation of risk factors for schizophrenia, those who met the criteria were included in the study. Next, a control group was recruited from two local educational institutions, matched by age, educational level, and gender, but excluding individuals with recurrent substance use or psychopathological indicators. Finally, participants diagnosed with schizophrenia were recruited through hospital services, a medium-stay psychiatric unit, and supervised housing facilities.

All assessments were carried out in the morning, in a quiet room, one-on-one, and were developed by a multidisciplinary team composed of a psychiatrist who assessed the PANSS, a psychologist who assessed the CAPE-42 and SCIP-S, and a speech therapist who assessed the BETA test.

### 2.4. Instruments

An ad hoc interview was conducted to gather data on sociodemographic variables. Additionally, the Drug Abuse Screening Test (DAST) [46,47] was administered to determine the type and frequency of drug use.

Participants in the PS group were assessed for psychopathological indices and predominant symptomatology using the Positive and Negative Syndrome Scale (PANSS) [39]. This semi-structured interview comprises 30 items, which are categorized into three factors: positive symptoms (7 items), negative symptoms (7 items), and general psychopathology (16 items). The scale yields positive, negative, general, and composite scores. The scale exhibits robust and reliable psychometric characteristics [48,49].

Furthermore, to standardize the various antipsychotic drugs and their prescribed dosages for these participants, the dose of antipsychotics taken by each participant was converted to its equivalent in chlorpromazine, in alignment with the recommendations of other studies [50,51]. Chlorpromazine was used as the reference antipsychotic because it is one of the most widely studied and its side effect profile is well established [52].

All participants in the HR group and the CG were administered the Community Assessment of Psychic Experience-42 (CAPE-42) questionnaire [53,54]. The aim of this instrument is to determine psychopathological symptomatology in the general population. It comprises 42 items assessing three dimensions of psychotic symptoms: positive (20 items), negative (14 items), and depressive (8 items). Each item is rated on a 4-point Likert scale ranging from ‘hardly ever’ (1) to ‘almost always’ (4). Participants who selected “sometimes”, “often”, or “always” are also asked to indicate how distressed they felt about the experience using a 4-point Likert scale. Scores of 2.8 in the positive dimension reflect indices that have a sensitivity of 83% with a positive predictive value of 74%, although they show a specificity of 49% and a negative predictive value of 63% according to the study of Mossaheb et al. [42].

To assess cognitive ability in all groups, we used the Cognitive Impairment in Psychiatry Screening Test (SCIP-S), this test is evaluated and validated on people with schizophrenia [55]. This test measures cognitive deficits in both the general population and people with mental health conditions. It consists of five subtests that examine immediate memory, working memory, verbal fluency, delayed memory, and processing speed, and provides an overall score. We used scores that had been transformed according to the relevant scale, for either a clinical or a community population. In addition to the psychometric properties, the Spanish translations are of excellent quality [56,57], as well as for other languages [58].

Finally, to measure semantic ability, we used the Aphasia Disorder Assessment Battery (BETA) [38]. This consists of 30 sheets, each with four pictures, and the respondent is required to identify the picture that is not semantically related to the others. For example, in an item showing a sofa, a chair, an armchair and a broom, the respondent is expected to select the broom as the item with no semantic association to the other three. In addition, the test has sensitivity values of 90.3% and specificity values of 98.7% [38].

### 2.5. Statistical Analysis

The data were analyzed using version 29.0 of the Statistical Package for the Social Sciences (SPSS^®^) (IBM^®^ Corp., Armonk, New York, NY, USA), with a significance level of 0.05. Prior to analyzing the results, the sample was tested for normality by the Kolmogorov–Smirnov test; none of the variables showed statistically significant differences (*p* > 0.05). Parametric analyses using ANOVA, ANCOVA, and bivariate Pearson correlation analyses were then performed to examine the relationship between the psychopathological and medication variables and the semantic test results. A chi-square statistical test was also performed for dichotomous variables.

## 3. Results

### 3.1. Demographic Characteristics

Table 1 presents the descriptive statistics for age, gender, educational level, and history of mental disorder. The participants had a mean age of 28.82 years (SD = 13.20), with a range of ages from 15 to 68.

To determine whether the high-risk and control groups were truly equivalent and buffer the possible effect of some of these variables, we performed an analysis of the variables in both groups, finding the following: sex (χ^2^(1) = 10,048; *p* = 0.546), age (F = 107,250, *p* = 0.360), educational level (χ^2^(2) = 0.611; *p* = 0.737). No effect was revealed for any of the variables analyzed, indicating that the groups were homogeneous.

### 3.2. Baseline Result

#### 3.2.1. Substance Use

As shown in Figure 1, the HR group has the highest rates of consumption, followed by the PS group and the CG. Alcohol is the most frequently consumed legal substance, while hashish is the most widely consumed illegal one.

#### 3.2.2. Cognitive Capacity

Table 2 shows the groups’ results on the SCIP-S subtests. The HR group had lower scores for verbal fluency, delayed memory, and processing speed. For the other subtests, the lowest scores corresponded to the PS group. There were no significant differences between the groups, except for in the working memory (*F* = 4.219 (DF: 2,1), *p* = 0.016) and immediate memory subtests (F = 4.528, (DF: 2,1) *p* = 0.012).

### 3.3. Psychopathology and Symptomatology

Table 3 and Table 4 show the PANSS positive, negative, and total scores, which are relatively similar and not particularly high, indicating the clinical stability of most participants. Notably, those experiencing negative symptoms had, on average, a higher chlorpromazine intake. Regarding the CAPE-42 score, the high-risk (HR) group had higher scores than the control group (CG), with significant differences observed only in positive symptoms and not in negative or depressive dimensions.

### 3.4. Semantic Associations

The descriptive statistics for the BETA test show that the PS group performed worst $\bar{x}$ = 23.97 (SD = 5.37), followed by the HR group $\bar{x}$ = 26.57 (SD = 2.29), with the CG achieving almost perfect results $\bar{x}$ = 28.77 (SD = 1.55). Additionally, statistically significant differences were found between the groups: F (2, 143) = 28.543; *p* ≤ 0.001; ηp2 = 0.374.

This finding was confirmed by an intergroup comparison, which revealed statistically significant differences between the groups (Table 5).

An ANCOVA was conducted to control for potential confounding variables, introducing medication, cognitive performance, age, sex, and educational level as covariates. The results showed that group differences on the BETA test remained statistically significant: *F* (4, 141) = 21.18, *p* < 0.001, η^2^ = 0.375. Among the covariates, medication (*F* (1, 141) = 12.89, *p* < 0.001, η^2^ = 0.084) and cognitive ability (*F* (1, 141) = 6.69, *p* = 0.002, η^2^ = 0.030) had a significant effect on BETA test scores. In both cases, higher levels of medication and lower cognitive performance were associated with poorer outcomes on the BETA test.

Table 6 shows the semantic association errors by group and item. Statistically significant differences were observed between the groups for all 30 items in the BETA test, with non-uniform error patterns.

We found items 28, 11, 9, and 3 particularly relevant in the schizophrenia group, where the highest error percentages were recorded, followed by the high-risk group.

### 3.5. Correlations Between Variables

Table 7 shows the statistically significant correlations between the BETA test and some of the study variables. For instance, it shows whether psychopathology or pharmacological factors influence the results of semantic association tests. We found that higher scores on medication, PANSS, and CAPE-42 correlate with worse scores on the BETA test, and we also observed that a higher cognitive score implies a better score on the BETA test.

## 4. Discussion

The aim of this study was to confirm the presence of semantic processing deficits in individuals with schizophrenia and to determine whether such deficits are also present in individuals at high risk of developing the disorder, with a view to their potential use as biomarkers. Evaluation using the BETA test [38] confirmed that the SP obtained the lowest scores, followed by the HR, and finally the CG. Furthermore, the differences between the groups were statistically significant and allowed clear discrimination among them.

In our study, the sociodemographic data are consistent with those in the literature. They disprove the idea that most patients with schizophrenia are men who are on multiple medications, have a history of mental health disorders, and have achieved low levels of academic attainment [59].

Data on substance use shows that high-risk participants report higher levels of consumption of all substances, particularly hashish. There is a well-established association between hashish use and an increased risk of developing serious mental health conditions such as schizophrenia [60,61].

Our data on psychopathology in the PS group, as measured via the PANSS, show that positive and negative symptoms are balanced and stabilized, given that all participants are receiving pharmacological treatment. These findings are consistent with those of other studies [62,63].

As explained, regarding medication, we used the dose equivalence proposed by Leucht et al. [56,57] to measure the total amount of pharmacological therapy administered to people with schizophrenia. The data show that those experiencing negative symptoms require an average higher dose of chlorpromazine. This is because such symptoms are more difficult to treat, resulting in poorer functionality and prognosis, and meaning that larger doses of medication are required [21,64].

As expected, the scores are higher in the group at risk of schizophrenia than in the control group. This suggests that the criteria proposed by Fonseca et al. [65] and Mossaheb et al. [42] effectively distinguish individuals at high risk of schizophrenia from those at lower risk. Although the use of these inclusion and exclusion criteria ensures that one group presents high-risk symptoms while the other does not, it limits the ability to fully assess the test’s sensitivity and specificity. Nevertheless, the instrument has demonstrated consistent and satisfactory performance across several languages [26,66,67,68]. Incorporating an objective and straightforward measure, such as the semantic relations test, alongside the self-report instrument, could therefore enhance risk detection and contribute to identifying a promising biomarker for schizophrenia. Future research should continue to examine the concurrent validity of the CAPE-42 in comparison with other psychopathological assessments, such as those described by Mossaheb et al. [42].

Based on these results, we can confirm that our sample comprises individuals in the prodromal stage of schizophrenia, particularly in the initial phase, when basic and non-specific symptoms are most prevalent. This is evidenced by the presence of positive, negative, and depressive symptoms that, while subclinical, contribute to functional impairment and psychological distress. These features effectively differentiate the high-risk group from the control groups [25,69,70].

The results for cognitive capacity, as measured by the SCIP-S, suggest that people with schizophrenia demonstrate fewer deficits than the control group (CG). In contrast, the high-risk (HR) group exhibits deficits, particularly in verbal fluency, delayed memory, and processing speed, when compared with the CG [71].

This could be explained by two main factors: in the high-risk group, repeated substance abuse impacts cognitive abilities [71], and/or, in some cases, these individuals of schizophrenia group would benefit from starting pharmacological treatment. Medication reduces the hyperactivation associated with the thought disorder, which would improve cognitive abilities [72], although some studies report a slowing of these abilities due to medication [73], as can be seen in the results of processing speed.

Regarding the participants’ data on semantic associations, as measured by the BETA test, significant differences were observed between the three groups. The SP group obtained the lowest scores, followed by the HR group, and finally the CG. These results are consistent with those of previous studies reporting semantic deficits in individuals with schizophrenia [13,74,75] and in individuals at high risk [76].

When considering possible reasons for these discrepancies, age and level of education could be seen as explanations, given that the SP group has the highest mean age. However, studies such as that by Gudayol-Ferré et al. [77] highlight that discrepancies in semantic results are generated not by age, but by having less than six years of education. Therefore, age may not be the underlying reason for our results. Furthermore, the HR group and the CG were matched in the studies, ruling out educational level as the cause of discrepancies between the groups.

Another possible explanation could be substance use, given that it causes cognitive impairment [71]. However, this hypothesis was dismissed because people with schizophrenia obtained the worst scores in semantic association tests, despite not having the highest rates of substance consumption. Furthermore, not all participants with recurrent substance use showed extreme deficits on the BETA test. Furthermore, these deficits were not reflected in most items, unlike the results in the PS group.

Consequently, our data could confirm the presence of specific semantic deficits in individuals with schizophrenia and those at high risk of developing the condition. In the latter group, however, such deficits appear to be more moderate [14,74,78]. However, future research should aim to develop novel methodological approaches to establish the causal nature of these findings, as the present data only allow for the identification of associations.

The analysis of errors in individual items reveals differences in the pattern of semantic association deficits between the PS and HR groups. According to semantic processing models, word retrieval begins with the activation of a set of semantic features connected to multiple words. These features activate words according to the strength and number of their connections. One word is then selected as the final response, while other activations are inhibited to ensure that retrieval is accurate [79,80].

The design of the BETA test involves selecting, from among four visual stimuli, those that share the highest number of categories, which requires semantic activation and inhibition abilities [38]. These processes may be impaired in individuals with schizophrenia [14] and, according to various studies, in people with addictions [81].

It can be observed that all three groups made more mistakes on items 9, 11, 21, 22, 26, 27, 28, and 29. Those with schizophrenia had the highest error rates. This can be explained by the concept of semantic interference [82,83]. This occurs when the activation of a word triggers the activation of semantic competitors that require inhibition, which can slow down the decision-making process and increase the probability of making errors.

These items may be more difficult due to the similarity between the correct image and the distractors. As Wienrich et al. [84] noted, when distractors are more similar, participants take longer to respond and show behavioral signs like longer fixations and repeated looks, which may reflect interference. This effect could be stronger in people with schizophrenia due to their greater cortical disinhibition [81,85,86], and might also affect high-risk individuals, possibly due to substance use [81].

In contrast, in items like 5, 10, 14, 18, 20, 23, and 24, where the distractors are less similar and the semantic interference is lower, errors are seen only in the schizophrenia group.

These findings suggest that some images demand more complex processing, which seems especially difficult for people with schizophrenia and, to a lesser extent, for those at high risk [14,81,82,84]. Simpler items, however, mainly challenge only the schizophrenia group. Future studies could explore this in more depth.

If these results were confirmed, the BETA test could be used to identify individuals at risk of developing schizophrenia, since the control group in this study answered more than 98% of the items correctly [38,87]. Various studies have corroborated the potential of language as an early biomarker of schizophrenia [76,88].

As expected, individuals with schizophrenia presented a higher overall PANSS score, implying a greater presence of psychopathological symptoms and a lower degree of semantic processing. This shows that poorer mental health leads to weaker semantic association abilities [78,89].

It is worth noting the correlation between increased psychopathological symptoms, as measured by the CAPE-42 test, and lower semantic memory scores in both the high-risk and control groups. Dopaminergic hyperactivation associated with psychopathology, specifically in cortical language areas such as the temporal lobe, could be the cause of these deficits [24,90].

These findings suggest that semantic deficits may manifest before the onset of schizophrenia [53,54] and are closely linked to psychopathological disorders. This positions them as possible early indicators for use in psychiatric services, suggesting that language may be a biomarker of schizophrenia and an explanatory factor [76].

The correlation between the dosage of chlorpromazine prescribed to people with schizophrenia and their performance in semantic tests may be explained by the fact that those with more severe symptoms are usually given higher doses of antipsychotic medication, which could affect their performance in the BETA test [51,91].

In our study, we found a correlation between the three groups’ cognitive capacity and the results of the semantic test. Those with higher cognitive skills scored higher on the BETA test [14,72].

Scientific evidence has indicated the existence of semantic deficits in people with schizophrenia and at high risk of developing it; however, until now, standardized tests were lacking to allow for clinical confirmation of these deficits. Our results, obtained through the BETA test, highlight these deficits, as this test allows us to detect linguistic alterations and, specifically, how people with schizophrenia or at high risk interpret meanings in their daily lives. This would support the clinical applicability and ecological validity of the BETA test.

Finally, the present research is not without limitations, such as the fact that the individuals in the schizophrenia group could not be matched in terms of educational level to the other groups in the study. The small sample size of the high-risk group is also a limitation, which may affect the statistical power of some analyses. Furthermore, it would have been interesting to analyze the effects of each type of antipsychotic drug taken by the participants in the PS group on their results in the semantic test, as well as to evaluate the associations between these drugs and formal thought disorders. Another important limitation is that the study design does not allow for the establishment of causal relationships, only relationships between the different variables. Future studies should therefore focus on this aspect.

## 5. Conclusions

The BETA test could be effective in assessing semantic deficits in patients with schizophrenia and in detecting such deficits in individuals at high risk of developing the condition. However, future studies will need to replicate these results in larger and more diverse samples.

Additionally, analyzing the errors could provide insight into semantic degradation, which appears to be more prevalent in individuals with higher psychopathological indices. This degradation begins with the most semantically demanding concepts and those with a greater number of competitors.

We consider it extremely important to conduct longitudinal studies of people at high risk, in order to confirm whether people with greater semantic and psychopathological deficiencies end up developing schizophrenia.

Therefore, semantic processing, particularly at more complex and abstract levels, may have the potential to identify individuals in the prodromal stages of schizophrenia. If these findings are replicated in future studies, including those conducted in other languages, semantic processing could be considered a potential biomarker for schizophrenia.

Currently, identifying individuals at high risk often relies on time-consuming assessments conducted by highly experienced clinicians. As our findings suggest, combining self-reported psychopathological measures with objective tests of semantic language processing could yield a protocol that is both sensitive and reliable for identifying individuals in high-risk states. However, further research is needed to validate and refine this approach.

## Figures and Tables

**Figure 1 healthcare-13-01958-f001:**
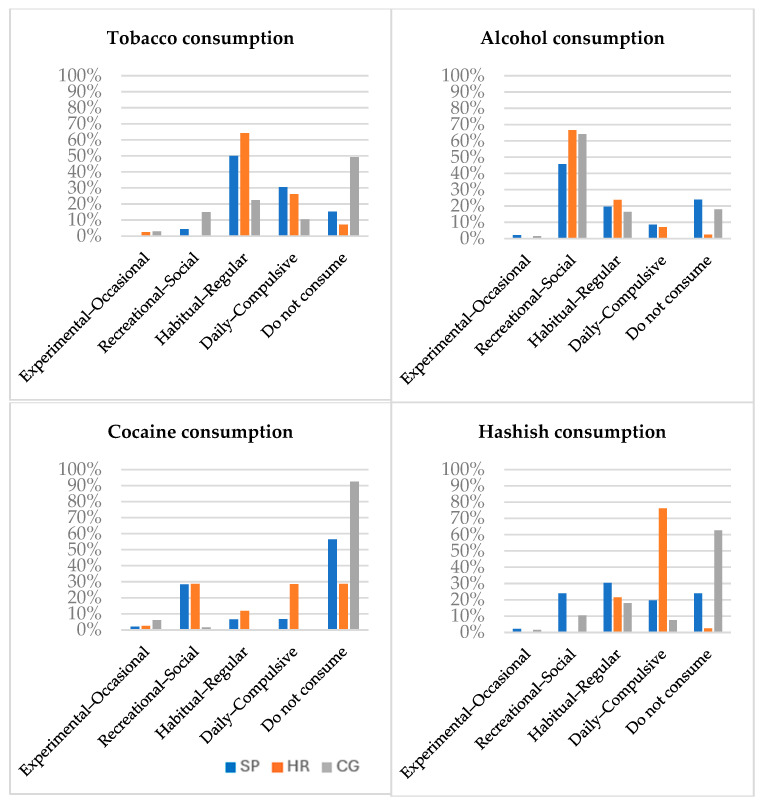
Types of consumption by substance, frequency and group. Note: PS = patients with schizophrenia; HR = individual at high risk; CG = control group.

**Table 1 healthcare-13-01958-t001:** Descriptive statistics for age, gender, level of education, and history of mental disorder.

		TS	SP	HR	CG
Age	Variables	146	46	40	60
	ℜ	28.82	45.17	22.57	20.45
	SD	13.20	11.58	4.71	2.25
	Minimum	15	24	15	15
	Maximum	68	68	33	33
Gender	Male	90 (61.6%)	30 (65.2%)	30 (75%)	30 (50%)
	Female	56 (38.4%)	16 (34.8%)	10 (25%)	30 (50%)
Level of education	Primary	12 (7.7%)	10 (21.7%)	1 (2.5%)	1 (1.7%)
	Secondary	124 (84.9%)	34 (73.9%)	35 (87.5%)	55 (91.7%)
	Higher Ed.	10 (6.8%)	2 (4.3%)	4 (10%)	4 (6.7%)
History of mental disorder in first-degree relatives	Yes	76 (52.1%)	36 (78.3%)	22 (55%)	18 (30%)
	No	70 (47.9%)	10 (21.7%)	18 (45%)	42 (70%)

Note: TS = total sample; SP = patients with schizophrenia; HR = individual at high risk; CG = control group; ℜ = average value of a set of numbers, calculated by adding all the values and dividing by the number of values; SD = dispersion of values in a set of data with respect to the mean.

**Table 2 healthcare-13-01958-t002:** Descriptive statistics for cognitive assessment.

	Group	ℜ	SD	Min	Mx	F	DF	*p*	ηp2
Total SCIP-S	PS	42.58	10.64	27	66	3.007	2	0.052	0.038
	HR	40.33	10.74	27	73				
	CG	45.01	8.52	27	61				
Immediate memory	PS	34.97	9.19	27	69	4.219	2	0.016	0.053
	HR	37.35	9.29	27	61				
	CG	39.74	7.72	27	61				
Working memory	SP	47.15	9.33	27	69	0.531	2	0.589	0.007
	HR	48.14	10.22	32	68				
	CG	46.28	8.42	32	71				
Verbal fluency	PS	52.60	11.97	27	71	1.977	2	0.142	0.025
	HR	50.45	10.96	27	71				
	CG	54.49	8.66	35	73				
Delayed memory	PS	42.56	9.19	27	60	4.528	2	0.012	0.056
	HR	38.69	8.02	27	58				
	CG	44.11	9.90	27	62				
Processing speed	PS	44.84	9.48	27	60	0.269	2	0.765	0.004
	HR	45.11	8	27	69				
	CG	45.91	6.91	27	61				

Note: PS = patients with schizophrenia; HR = individual at high risk; CG = control group; ℜ = average value of a set of numbers, calculated by adding all the values and dividing by the number of values; SD = dispersion of values in a set of data with respect to the mean. F = value of the ANOVA; DF = degrees of freedom; *p* = statistically significant values; ηp2 = potential eta squared of the ANOVA.

**Table 3 healthcare-13-01958-t003:** Descriptive statistics for psychopathology in the clinical group.

	PS (N = 46)		
	ℜ (SD)	Min.	Max.
PANSS Negative	19.13 (9.98)	8	48
PANSS Positive	19.67 (7.29)	8	44
PANSS Overall	49.73(12.30)	30	86
Chlorpromazine	433.71(258.28)	10.23	1040.67
Chlorpromazine PS with positive symptoms	421.27 (260.27)	10.23	1040.67
Chlorpromazine PS with negative symptoms	451.38 (261.47)	10.23	950

Note: N= Sample; PS = patients with schizophrenia; **ℜ** = average value of a set of numbers, calculated by adding all the values and dividing by the number of values; SD: dispersion of values in a set of data with respect to the mean.

**Table 4 healthcare-13-01958-t004:** Descriptive statistics for psychopathological indices and symptomatology in community groups.

CAPE 42		HR (N = 40)			CG (N= 60)			Student’s t	
		ℜ (SD)	Min.	Max.	ℜ (SD)	Min.	Max.	F	Sig
Positive dimension	Positive symptoms	35.7 (8.12)	23	66	25.3 (3.61)	15	32	16.936	<0.001
	Weighted positive symptoms	1.78 (0.40)	1.6	3.30	1.26 (0.18)	0.75	1.6	16.936	<0.001
	Positive distress	21.67 (9.8)	10	50	7.76 (4.42)	2	23	41.959	<0.001
	Weighted positive distress	1.78 (0.40)	1.50	2.50	0.38 (0.22)	0.10	1.15	41.959	<0.001
	Total weighted positive score	2.87 (0.82)	2.8	5.30	1.65 (0.35)	1.10	2.60	32.494	<0.001
Negative dimension	Negative symptoms	26.9 (7.08)	19	48	21.3 (4.03)	13	27	6.430	0.013
	Weighted negative symptoms	1.92 (0.50)	1.36	3.43	1.52 (0.28)	0.93	1.8	6.430	0.013
	Negative distress	20.97 (8.34)	13	53	10.68 (5.75)	0	23	1.372	0.244
	Weighted negative distress	1.49 (0.59)	1.2	3.79	0.76 (0.41)	0	1.64	1.372	0.244
	Total weighted negative score	3.42 (1.02)	2.43	7	2.28 (0.67)	1	3.36	2.269	0.136
Depressive dimension	Depressive symptoms	17.97 (5.2)	10	30	14 (2.97)	9	19	18.315	<0.001
	Weighted depressive symptoms	2.24 (0.65)	1.25	3.75	1.75 (0.37)	1.13	2.2	18.315	<0.001
	Depressive distress	15.2 (5.02)	9	28	9.66 (4.42)	1	22	1.747	0.189
	Weighted depressive distress	1.90 (0.62)	1.13	3.50	1.20 (0.55)	0.13	2.4	1.747	0.189
	Total weighted depressive score	4.15 (1.14)	2.50	6.75	2.95 (0.86)	1.25	4.5	5.604	0.020
Total dimension	Total symptoms	81.55(18.44)	62	143	60.5 (7.3)	47	62	32.819	<0.001
	Weighted total symptoms	1.94 (0.43)	1.45	3.40	1.44 (0.17)	1.12	1.69	32.819	<0.001
	Total distress	59.05(19.7)	35	117	28.28 (11.1)	5	52	9.687	0.002
	Weighted total distress	1.40 (0.46)	1.3	2.79	0.67 (0.26)	0.12	1.24	9.687	0.002
	Total CAPE-42 weighted score	3.34 (0.85)	2.45	5.55	2.11 (0.41)	1.31	2.4	21.388	<0.001

Note: HR =N=Sample; individual at high risk; CG = control group; **ℜ** = average value of a set of numbers, calculated by adding all the values and dividing by the number of values; SD: dispersion of values in a set of data with respect to the mean. F is the value of Student's t test between the two groups on the CAPE-42 questionnaire: Sig = 0.005.

**Table 5 healthcare-13-01958-t005:** Comparison of means in the ANOVA by group in the BETA test.

	Intergroups Comparison	Difference in Means	CI 95%		*p*
			Low.	High.	
BETA test Groups	HR VS PS	2.59 *****	1.19	3.99	<0.001
	HR VS CG	−2.20 *****	−3.49	−0.91	<0.001
	PS VS CG	−4.79 *****	−6.05	−3.54	<0.001

Note: PS = patients with schizophrenia; HR = individual at high risk; CG = control group; *p* = statistical significance values control group; *p* = statistically significant; * = below 0.05.

**Table 6 healthcare-13-01958-t006:** Analysis by group of semantic association errors made in the BETA test.

	PS		HR		CG		X	* p *
	Failed	success	Failed	success	Failed	success		
Item 3	11 (23.9%)	35 (76.1%)	12 (30%)	28 (70%)	7 (11.7%)	53 (88.3%)	6304	0.043
Item 4	5 (10.9%)	41 (89.1%)	1 (2.5%)	39 (97.5%)	1 (1.7%)	59 (98.3%)	6.171	0.046
Item 5	4 (8.7%)	42 (91.3%)	0	40 (100%)	0	60 (100%)	9.729	0.008
Item 8	9 (19.6%)	37 (80.4%)	4 (10%)	38 (90%)	1 (1.7%)	59 (98.3%)	8.579	0.014
Item 9	18 (39.1%)	28 (60.9%)	18 (45%)	24 (55%)	7 (11.7%)	53 (88.3%)	14.086	<0.001
Item 10	5 (10.9%)	41 (89.1%)	0	40 (100%)	0	60 (100%)	6.657	0.036
Item 11	27 (58.7%)	19 (41.3%)	18 (45%)	22 (55%)	10 (16.7%)	50 (83.3%)	21.475	<0.001
Item 13	10 (21.7%)	36 (78.3%)	2 (5%)	38 (95%)	3 (5%)	57 (95%)	9.251	0.010
Item 14	4 (8.7%)	42 (91.3%)	0	40 (100%)	0	60 (100%)	9.729	0.008
Item 16	3 (6.5%)	43 (93.5%)	1 (2.5%)	39 (97.5%)	0	60 (100%)	12.512	0.002
Item 17	6 (13%)	40 (87%)	1 (2.5%)	39 (97.5%)	0	60 (100%)	11.370	0.003
Item 18	5 (10.9%)	41 (89.1%)	3 (7.5%)	37 (92.5%)	0	60 (100%)	7.045	0.030
Item 19	4 (8.7%)	42 (91.3%)	0	40 (100%)	0	60 (100%)	9.729	0.008
Item 20	3 (6.5%)	43 (93.5%)	0	40 (100%)	0	60 (100%)	7.249	0.027
Item 21	9 (19.6%)	37 (80.4%)	0	40 (100%)	0	60 (100%)	22.641	<0.001
Item 22	17 (37%)	29 (63%)	6 (15%)	34 (85%)	3 (5%)	57 (95%)	20.023	<0.001
Item 23	4 (8.7%)	42 (91.3%)	0	40 (100%)	0	60 (100%)	9.729	0.008
Item 24	3 (6.5%)	43 (93.5%)	0	40 (100%)	0	60 (100%)	7.249	0.027
Item 25	7 (15.2%)	39 (84.8%)	1 (2.5%)	39 (97.5%)	1 (1.7%)	59 (98.3%)	8.344	0.015
Item 26	16 (34.8%)	30 (65.2%)	12 (30%)	30 (70%)	1 (1.7%)	59 (98.3%)	20,803	<0.001
Item 27	16 (34.8%)	30 (65.2%)	3 (7.5%)	39 (92.5%)	0	60 (100%)	32.081	<0.001
Item 28	36 (78.3%)	10 (21.7%)	25 (62.5%)	15 (37.5%)	11 (18.3%)	49 (81.7%)	41.007	<0.001
Item 29	16 (34.8%)	30 (65.2%)	3 (7.5%)	39 (92.5%)	3 (5%)	57 (95%)	22.919	<0.001
Item 30	10 (21.7%)	36 (78.3%)	5 (12.5%)	37 (87.5%)	1 (1.7%)	59 (98.3%)	12.234	0.002

Note: PS = patients with schizophrenia; HR = individual at high risk; CG = control group; X = chi-square test; *p* = statistically significant.

**Table 7 healthcare-13-01958-t007:** Correlations between the groups and overall BETA test scores.

		Amount of Chlorpromazine PS	PANSS General PS	CAPE-42 Total HR Persons and CG	CAPE 42 Distress HR Persons and CG	SCIP Cognitive Results PS HR and CG
BETA	Pearson’s correlation	−0.342 *****	−0.643 ******	−0.566 ******	−0.579 ******	0.259 ******
	Sig. (Bilateral)	0.020	<0.001	<0.001	<0.001	0.001

Note: PS = patients with schizophrenia; HR = individual at high risk; CG = control group; * = below 0.05; ** = below 0.01.

## Data Availability

The original contributions presented in this study are included in the article. Further inquiries can be directed to the corresponding author.

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
