# Peer review of "Semantic Processing Deficits and Their Use as Early Biomarkers in Schizophrenia"

_healthcare, 2025, doi:10.3390/healthcare13161958_

Round 1

Reviewer 1 Report

Comments and Suggestions for Authors

On page 3/line 99 you state you are examining people with schizophrenia though earlier you state it is at-risk groups you are examining. Please clarify how the risk groups fit into hypotheses. 

Page 4/line 131 you state participants with co-occurring presentations were excluded. How many were these? How were these screened? It is difficulty to imagine a large sample without trauma, substance use, mood component, etc. 

The discussion starts with an overview of demographics - this is odd given that was not a primary focus of the paper. I suggest re-organizing.  

Author Response

Dear Reviewer 1,

We would like to acknowledge the comprehensive review of our work. We also thank you for the opportunity to resubmit them.

Please find attached point-by-point responses, which are coloured in red. All the comments that required a change in the manuscript have been included in the main document and are also highlighted.

We believe that the manuscript has been substantially improved, and we hope that it can be considered for publication shortly after your reassessment.

Best regards,

Begoña Polonio-López, on behalf of all the authors

Reviewer 2 Report

Comments and Suggestions for Authors

This study tackles an important and tricky problem: how hard it is to detect schizophrenia early. The authors laid out their research question in an engaging way in the introduction, and clearly spelled out the study’s goals by the end of that section. The paper itself is written in a clear and easy-to-follow style.

To help strengthen the study and highlight the value of its findings, here are my suggestions:

  1. In line 13, the authors mention excluding some participants due to certain psychopathological signs, but these weren’t explained or defined earlier. That’s confusing. Also, why weren’t participants screened before being included in the study?
  2. In Table 1, it says “history of mental disorder.” What exactly does that mean, especially for those in the schizophrenia group?
  3. The paper doesn’t explain how participants were recruited. Without this, it's hard to rule out selection bias or other biases. It would be helpful if the authors shared their recruitment strategy and how they formed the study groups.
  4. In line 164, starting a new paragraph gives the impression of a new topic, but it’s actually continuing the description of the tool from the previous sentence. It may be clearer to keep this together.
  5. The results section begins with drug abuse data, which wasn’t one of the study’s objectives. Also, the baseline characteristics are placed under methods instead of results, which is confusing.
  6. The authors ran a large number of statistical tests. It's generally a good idea to adjust for multiple testing to avoid drawing false conclusions from random findings.
  7. In line 264, the authors claim their results challenge commonly known demographic features of people with schizophrenia. But without a well-described recruitment strategy, this could be due to selection bias. They should clarify how their study supports that claim.
  8. Similar concerns apply to the discussion on Hashish use. The authors compare their findings to other studies with different designs, which might not be a fair comparison.
  9. In line 296, the authors mention that people with schizophrenia in their study showed fewer cognitive issues than other groups. They suggest treatment might have improved their cognitive function beyond normal levels. That’s a strong claim—can they explain it more clearly?
  10. The conclusions about the CAP42 scale seem a bit exaggerated. Since the study included patients based on this scale, it’s hard to use the findings to assess how well the scale actually distinguishes between groups.
  11. The discussion section leans heavily on causal explanations, but this is a cross-sectional study—it can show associations, not cause-and-effect. The discussion of the BETA score should be reworded, and it would be great to suggest future research to explore semantic tests utility for early schizophrenia detection.
  12. The limitations section is quite reduced. It doesn’t mention the biggest drawback—the study design itself.
  13. The methods say the variables were tested for normality. It’s important for the authors to include what did the tests show about this.

Author Response

Dear Reviewer 2,

We would like to acknowledge the comprehensive review of our work. We also thank you for the opportunity to resubmit them.

Please find attached point-by-point responses, which are coloured in red. All the comments that required a change in the manuscript have been included in the main document and are also highlighted.

We believe that the manuscript has been substantially improved, and we hope that it can be considered for publication shortly after your reassessment.

Best regards,

Begoña Polonio-López, on behalf of all the authors

Reviewer 3 Report

Comments and Suggestions for Authors

I appreciate the opportunity to review this manuscript, which addresses a clinically and theoretically relevant topic with originality and scientific rigor. I believe the study makes a valuable contribution to the field of psychopathology and the early assessment of psychosis risk.

I have prepared a detailed report with comments and suggestions aimed at improving the clarity of the presentation, strengthening certain methodological justifications, and optimizing the presentation of the results. This report is attached to the present review.

I hope these suggestions will be helpful in the revision process of the manuscript.

Comments on the Quality of English Language

The level of English is adequate; however, a final language review is recommended to correct minor errors and improve overall fluency.

Author Response

Dear Reviewer 3,

We would like to acknowledge the comprehensive review of our work. We also thank you for the opportunity to resubmit them.

Please find attached point-by-point responses, which are coloured in red. All the comments that required a change in the manuscript have been included in the main document and are also highlighted.

We believe that the manuscript has been substantially improved, and we hope that it can be considered for publication shortly after your reassessment.

Best regards,

Begoña Polonio-López, on behalf of all the authors

Round 2

Reviewer 2 Report

Comments and Suggestions for Authors

the text is improved despite some persistent limitations in terms of recruitment strategy development and exploitation of the results and their interpretation.

Author Response

Dear Reviewer 2,

Thank you for your thoughtful feedback. After carefully reviewing both your initial comments and your most recent observations, we have made several changes to the manuscript, particularly in the Methodology, Procedure, and Conclusions sections.

Please find attached point-by-point responses, which are colored in red.

We greatly appreciate your insights and believe these revisions have enhanced the clarity and rigor of the manuscript.

Best regards,

Begoña Polonio-López, on behalf of all the authors
